# Comparison of Multiparametric MRI, [^68^Ga]Ga-PSMA-11 PET-CT, and Clinical Nomograms for Primary T and N Staging of Intermediate-to-High-Risk Prostate Cancer

**DOI:** 10.3390/cancers15245838

**Published:** 2023-12-14

**Authors:** Omar Marek Tayara, Kacper Pełka, Jolanta Kunikowska, Wojciech Malewski, Katarzyna Sklinda, Hubert Kamecki, Sławomir Poletajew, Piotr Kryst, Łukasz Nyk

**Affiliations:** 1Second Department of Urology, Centre of Postgraduate Medical Education, 01-813 Warsaw, Poland; wojtek.malewski@gmail.com (W.M.); slawomir.poletajew@gmail.com (S.P.); piotr.kryst@cmkp.edu.pl (P.K.); ukinyk@poczta.fm (Ł.N.); 2Department of Nuclear Medicine, Medical University of Warsaw, 02-097 Warsaw, Poland; kacper.pelka@wum.edu.pl (K.P.); jolanta.kunikowska@wum.edu.pl (J.K.); 3Department of Methodology, Laboratory of Centre for Preclinical Research, Medical University of Warsaw, 02-091 Warsaw, Poland; 4Department of Radiology, Centre of Postgraduate Medical Education, 01-809 Warsaw, Poland; 5Diagnostic Radiology Department, Central Clinical Hospital of the Ministry of the Interior in Warsaw, 02-507 Warsaw, Poland

**Keywords:** PSMA, [^68^Ga]Ga-PSMA-11, prostate cancer, nomograms, PET-CT

## Abstract

**Simple Summary:**

Prostate cancer is the second most common cancer among men worldwide. A sensitive pre-operative diagnosis of prostate cancer is extremely important because it allows the best treatment path to be chosen for the patient. This includes imaging examinations such as magnetic resonance imaging, computed tomography, or positron emission tomography with a prostate-specific membrane antigen. In clinical practice, nomograms can also be based on the patient’s results, which inform about the potential extent of the disease but have no possibility of determining the location of suspected metastatic lesions. The aim of this prospective study is to analyze the combination of magnetic resonance and positron-computed tomography against available nomograms to determine sensitivities, specificities, and diagnostic possibilities. The results indicate the superiority of combined imaging data relative to nomograms in the diagnosis of local advancement and lymph node involvement. Magnetic resonance imaging was the best for extra-prostatic extension and seminal vesicle involvement. Positron emission tomography with a prostate-specific membrane antigen was the most sensitive to assess lymph node involvement.

**Abstract:**

Purpose of the Report: Although multiparametric magnetic resonance imaging (mpMRI) is commonly used for the primary staging of prostate cancer, it may miss non-enlarged metastatic lymph nodes. Positron emission tomography-computed tomography targeting the prostate-specific membrane antigen (PSMA PET-CT) is a promising method to detect non-enlarged metastatic lymph nodes, but more data are needed. Materials and Methods: In this single-center, prospective study, we enrolled patients with intermediate-to-high-risk prostate cancer scheduled for radical prostatectomy with pelvic node dissection. Before surgery, prostate imaging with mpMRI and PSMA PET-CT was used to assess lymph node involvement (LNI), extra-prostatic extension (EPE), and seminal vesicle involvement (SVI). Additionally, we used clinical nomograms to estimate the risk of these three outcomes. Results: Of the 74 patients included, 61 (82%) had high-risk prostate cancer, and the rest had intermediate-risk cancer. Histopathology revealed LNI in 20 (27%) patients, SVI in 26 (35%), and EPE in 52 (70%). PSMA PET-CT performed better than mpMRI at detecting LNI (area under the curve (AUC, 95% confidence interval): 0.779 (0.665–0.893) vs. 0.655 (0.529–0.780)), but mpMRI was better at detecting SVI (AUC: 0.775 (0.672–0.878) vs. 0.585 (0.473–0.698)). The MSKCC nomogram performed well at detecting both LNI (AUC: 0.799 (0.680–0.918)) and SVI (0.772 (0.659–0.885)). However, when the nomogram was used to derive binary diagnoses, decision curve analyses showed that the MSKCC nomogram provided less net benefit than mpMRI and PSMA PET-CT for detecting SVI and LNI, respectively. Conclusions: mpMRI and [^68^Ga]Ga-PSMA-11 PET-CT are complementary techniques to be used in conjunction for the primary T and N staging of prostate cancer.

## 1. Introduction

The primary staging of prostate cancer is crucial in patients considered for radical prostatectomy, with the T stage guiding the scope of local tumor resection, and the risk of nodal involvement justifying the extent of pelvic lymph node dissection [1]. According to current guidelines, nerve-sparing radical prostatectomy cannot be offered to patients with an extra-prostatic extension (EPE and cT3a) or seminal vesicle involvement (SVI and cT3b) [2], conditions best identified through multiparametric MRI (mpMRI). An extended pelvic lymph node dissection, indicated in patients with high-risk cancer and a risk of nodal metastases greater than 7% [1,3], is often determined based on imaging findings.

Staging nomograms, while used in clinical practice to gauge the risks of EPE, SVI, and lymph node involvement (LNI), provide only estimated probabilities [4,5]. These nomograms, however, do not replace the need for precise imaging techniques, as they lack the exact anatomical details necessary for surgical planning. Therefore, prostate imaging, particularly mpMRI, is necessary in all patients considered for radical prostatectomy.

Routine abdominopelvic imaging with computed tomography (CT) or magnetic resonance imaging (MRI) is deemed insufficient for the primary staging of prostate cancer. Currently, mpMRI is regarded as the best method for the T staging of prostate cancer, displaying high accuracy for diagnosing both EPE [6,7,8,9,10,11] and SVI [12,13,14]. However, mpMRI is less accurate for assessing LNI because it may miss non-enlarged metastatic lymph nodes [15]. This limitation necessitates complementary imaging techniques, such as positron emission tomography-computed tomography targeting the prostate-specific membrane antigen (PSMA PET-CT), which is emerging as a potentially more effective method than mpMRI for detecting LNI in patients with intermediate-to-high-risk prostate cancer [16,17,18,19,20], though further data are needed to confirm this. PSMA is a transmembrane protein expressed by prostate cells, with its expression increasing with greater cell dysplasia [1].

In our prospective study, we compared the performance of selected clinical nomograms, mpMRI, and [^68^Ga]Ga-PSMA-11 PET-CT for diagnosing LNI, SVI, and EPE, aligning our research with these evolving guidelines and the quest for more accurate staging methods.

## 2. Methods

### 2.1. Study Setting and Patients

This prospective study was carried out between March 2020 and September 2022. We included patients with newly diagnosed, biopsy-proven, treatment-naive, high-risk prostate cancer or with intermediate-risk prostate cancer and a risk of nodal involvement of at least 7% on the Briganti nomogram [1]. After the initial staging of CT scans and bone scans, we excluded patients with high-volume disease according to the CHAARTED criteria [21], as well as those who did not agree to radical prostatectomy. We performed robotic-assisted or laparoscopic radical prostatectomy with pelvic lymph node dissection including the external, hypogastric, and common iliac chain bilaterally and obturator lymph nodes, as recommended by the European Association of Urology [1].

The study was approved by the local Bioethics Committee of Centre of Postgraduate Medical Education (no. 47/2021). All patients signed informed consent before enrollment.

### 2.2. Clinical Staging and Risk Classification

Local tumor stage was ascertained by digital rectal examination. Biopsy specimens were evaluated by a routine pathological examination to obtain the International Society of Urological Pathology (ISUP) grades [22]. The Memorial Sloan Kettering Cancer Center (MSKCC, https://www.mskcc.org/, accessed on 1 April 2020) nomogram and Partin tables [5] were used to obtain the probabilities of LNI, SVI, and EPE. The probability of LNI was additionally assessed with the 2019 Briganti nomogram [23].

### 2.3. Multiparametric MRI Protocol

All images were acquired in line with the Prostate Imaging Reporting and Data System (PIRADS) v. 2. We used 1.5 T and 3.0 T scanners from different vendors (Siemens (Berlin, Gremany), Philips Healthcare (Amsterdam, The Netherlands), and General Electric (Boston, MA, USA)) with body phased-array coils. None of the patients were examined with endorectal coils. In all cases, the mpMRI protocol included sagittal T2WI, axial T2WI, diffusion-weighted imaging (DWI; b values of 0 and 1000, 2000 s/mm^2^), and dynamic contrast-enhanced (DCE) sequences. Apparent diffusion coefficient (ADC) maps were delivered along with the performed study. At least one sequence had to provide full coverage of the pelvis (large field of view). Because the assessment of pelvic lymph nodes is based on their size and shape, we considered one sequence with a large field of view sufficient for this purpose.

The ADC and T2WI images were evaluated by one radiologist (K.S., with 11 years of experience in prostate MRI), who was blinded to all data. Prostate volume was calculated as follows: length × width × height × 0.52. Lymph nodes were evaluated based on T2WI, DWI, and DCE images of both large and small fields of view. The criteria for LNI were as follows: threshold of short axis diameter, DWI restriction, shape of the lymph node, and contrast enhancement [24,25,26]. Figure 1 exemplifies how we classified LNI by mpMRI and PET-CT.

### 2.4. [^68^Ga]Ga-PSMA-11 PET-CT Protocol

Radiopharmaceutical preparation was performed as previously described [27]. [^68^Ga]Ga-PSMA-11 was intravenously injected 60 min before PET/CT acquisition. The injected activity was 2 MBq/kg of body mass (0.054 mCi/kg). Diuresis was induced with 20 mg of furosemide given intravenously just after the [^68^Ga]Ga-PSMA-11 injection to improve image quality. PET/CT was performed with the Biograph 64 TruePoint scanner (Siemens Medical Solutions Inc., Malvern, PA, USA) from the skull apex to the thighs, with a 3 min acquisition time per bed position (reconstruction: 3 iterations and 21 subsets). PET images were acquired directly after the voiding of the bladder. Non-enhanced CT was performed for attenuation correction and localization. PET-CT image analysis was performed using the Siemens Workstation (Syngovia, MMWS, Siemens Medical Solutions Inc., USA). In visual analysis, abnormal uptake was determined as a positive lesion when a lesion exhibited non-physiologically increased uptake that was discernible above background activity. The examination was assessed by two readers, who had at least 4 years of experience in the assessment of [^68^Ga]Ga-PSMA PET-CT; the result was a consensus between them.

### 2.5. Surgical Procedure

All procedures were performed through a six-port transperitoneal approach using the four-arm *da Vinci Surgical System*. After localization of the external iliac vessels, the peritoneum was incised, and lymph nodes were dissected, starting with the fibrofatty tissue along the external iliac vessels, with the distal limit being the deep circumflex vein and the femoral canal. The fibrofatty tissue within the obturator fossa was removed, and the obturator nerve was completely skeletonized. The lateral limit consisted of the pelvic sidewall, and, medially, the dissection limit was the perivesical fat. Proximally, lymph node dissection included removal of all lymph nodes along the common iliac vessels up to the aortic bifurcation. The lymph nodes located laterally and medially to the internal iliac artery were removed. Lastly, the ureters and the iliac vessels were completely skeletonized up to the aortic bifurcation. Each group of nodes was placed in a separate container with information for the pathologist to precisely determine the location and side of the changes.

### 2.6. Pathologic Examination

The resected specimens were assessed in a pathologic examination according to the current guidelines [22,28]. EPE was defined as pathologic stage 3a, whereas SVI was defined as pathologic stage 3b. All nodal specimens were prospectively mapped according to their anatomic location and sent for pathologic assessment. All pathologic analyses were performed by one genitourinary pathologist.

### 2.7. Statistical Analysis

Means (standard deviations), medians (interquartile ranges), and counts (percentages) were used as descriptive statistics. The descriptive statistics were calculated for the total sample and by the LNI status. Quantitative and categorical variables were compared using Wilcoxon rank sum test and Fisher’s exact or Pearson’s chi-squared test, respectively. Receiver operating characteristic (ROC) curve analyses were used to compare the performance of the MSKCC nomogram, Partin tables, 2019 Briganti nomogram, mpMRI, and [^68^Ga]Ga-PSMA-11 PET-CT at diagnosing LNI, SVI, and EPE. The post-operative pathologic examination was used as reference. The Youden statistic was used to calculate optimal cut-offs for the risk scores derived from the clinical nomograms. Sensitivities, specificities, positive prognostic values, and negative prognostic values were calculated accordingly.

We used decision curve analyses to compare the net benefit of diagnosing LNI and SVI based on mpMRI, [^68^Ga]Ga-PSMA-11 PET-CT, and the MSKCC nomogram, which had the best characteristics of all the clinical scores in our cohort (Section 3). In the decision curves analyses, the MSKCC diagnoses were binary (“yes” vs. “no”), which enabled direct comparisons with the imaging methods, which also provide binary classifications.

All calculations were completed in R software (version 4.2.2), with the pROC package used for ROC curve analyses [29], and the “dcurves” package for decision curve analyses.

## 3. Results

### 3.1. Characteristics of Patients

Of 93 patients assessed for eligibility, 7 were excluded due to high-volume disease and 12 due to a lack of consent to prostatectomy (Figure 2 shows patient selection). Of the 74 patients included, 9 (12%), 52 (82%), and 13 (18%) had metastatic, high-risk, and intermediate-risk prostate cancer, respectively. The tumors were palpable in 46 (62%) patients. Histopathology revealed LNI in 20 (27%) patients, SVI in 26 (35%), and EPE in 52 (70%). Distant metastases were found by [^68^Ga]Ga-PSMA-11 PET-CT in 9 (12%) patients. Compared to other patients, patients with LNI had significantly more advanced disease based on nearly all measures (see Table 1 for detailed characteristics).

### 3.2. ROC Curve Analyses

For predicting LNI, the MSKCC nomogram had the greatest AUC (0.799; 95% CI: 0.680–0.918) of all the methods; the AUC was similar for [^68^Ga]Ga-PSMA-11 PET-CT (0.779; 95% CI: 0.665–0.893) but lower for mpMRI (0.655; 95% CI: 0.529–0.780; see Figure 3A). The Briganti nomogram had the highest sensitivity (94.7%) but the lowest specificity (48.1%) of all the methods for predicting LNI. [^68^Ga]Ga-PSMA-11 PET-CT predicted LNI with a greater sensitivity (65%) and specificity (90.7%) compared with mpMRI (55% and 75.9%, respectively; see Figure 3A and Table 2 for details).

For predicting SVI, mpMRI had the greatest AUC (0.775; 95% CI: 0.672–0.878) of all the methods, but the AUC for the MSKCC nomogram was similar (0.772; 95% CI: 0.659–0.885; see Figure 3B). The sensitivities of mpMRI and the MSKCC nomogram were similar (65.4% vs. 69.2%), but mpMRI had a greater specificity (89.6% vs. 75%). [^68^Ga]Ga-PSMA-11 PET-CT had the lowest AUC (0.585; 95% CI: 0.473–0.698) and sensitivity (30%) of all the methods for predicting SVI (see Figure 3B and Table 2).

For predicting EPE, the AUCs for the MSKCC nomogram, Partin tables, and mpMRI were similar (between 0.566 and 0.613; see Figure 3C), with the clinical scores having higher sensitivity but lower specificity compared with mpMRI (see Figure 3C and Table 2).

Of all the clinical scores, the MSKCC nomogram had the best performance for predicting LNI, SVI, and EPE. Therefore, binary diagnoses based on the MSKCC nomogram were compared with those derived from mpMRI and [^68^Ga]Ga-PSMA-11 PET-CT by decision curve analyses.

### 3.3. Decision Curve Analyses

Decision curve analyses showed that [^68^Ga]Ga-PSMA-11 PET-CT had a greater net benefit than mpMRI did for predicting LNI across all threshold probabilities, with no additional benefit of combining [^68^Ga]Ga-PSMA-11 PET-CT with mpMRI (Figure 4A). In contrast, mpMRI had a greater net benefit across all threshold probabilities for predicting SVI than [^68^Ga]Ga-PSMA-11 PET-CT did, with no substantial benefit of combining mpMRI with [^68^Ga]Ga-PSMA-11 PET-CT (Figure 4B).

[^68^Ga]Ga-PSMA-11 PET-CT provided a greater benefit across all threshold probabilities than the MKSCC nomogram did at predicting LNI, with an additional benefit of combining these two methods across the greatest threshold probabilities (Figure 4C). mpMRI provided a greater net benefit than the MSKCC nomogram at detecting SVI, with an additional benefit of combining the two methods within the range of lower threshold probabilities (Figure 4D).

## 4. Discussion

The imaging methods for the primary staging of prostate cancer have improved over the past decades, and imaging is now indicated in all patients considered for radical prostatectomy [1]. Currently, mpMRI seems the best method for local T staging, whereas [^68^Ga]Ga-PSMA-11 PET-CT should also be used to improve the N staging [10]. Moreover, [^68^Ga]Ga-PSMA-11 PET-CT can detect distant metastases [30]. Numerous studies assessed the role of [^68^Ga]Ga-PSMA-11 PET-CT in the primary T and N staging of intermediate- or high-risk prostate cancer. Currently, its role in the diagnosis of a primary lesion is not established, as PSMA is not a simple-specific marker and can occur in benign mains like BPH or inflammatory lesions. For predicting LNI, the sensitivity of [^68^Ga]Ga-PSMA-11 PET-CT ranged between 33% and 100%, whereas specificity ranged from 80% to 100% [31]. Similarly, for assessing SVI and EPE, the sensitivity of [^68^Ga]Ga-PSMA-11 PET-CT widely ranged: from 11% to 94% for SVI, and from 0% to 94% for EPE [31]. Our results for the prediction of LNI by [^68^Ga]Ga-PSMA-11 PET-CT are within the range found in published work (sensitivity, 65%; specificity, 91%). We observed considerably lower sensitivity and specificity of [^68^Ga]Ga-PSMA-11 PET-CT for predicting SVI (see Table 2) compared with previous work [32]. As the role of [^68^Ga]Ga-PSMA-11 PET-CT in predicting EPE is not yet firmly established in the literature, we decided not to use [^68^Ga]Ga-PSMA-11 PET-CT for this purpose in our study [33,34].

Similar to our study, Ferraro et al. reported that [^68^Ga]Ga-PSMA-11 PET-CT had a sensitivity of 58% and a specificity of 98% for predicting LNI in patients with intermediate-to-high-risk prostate cancer [35]. Likewise, in another study among 61 patients with intermediate-to-high-risk prostate cancer, [^68^Ga]Ga-PSMA-11 PET-CT had a sensitivity of 67% and a specificity of 98% for detecting LNI [32].

In addition, investigators in that study were not able to use [^68^Ga]Ga-PSMA-11 PET-CT to reliably ascertain EPE but reported a greater sensitivity (58%) and specificity (98%) for detecting SVI than seen in our current study [32]. Gupta et al. found that [^68^Ga]Ga-PSMA-11 PET-CT displayed a sensitivity of 53% and a specificity of 99% for predicting LNI, whereas, for detecting SVI, the sensitivity was 55%, and the specificity was 100% [36]. In contrast to our study, those authors reported both the high sensitivity (63%) and specificity (100%) of [^68^Ga]Ga-PSMA-11 PET-CT for diagnosing EPE [36]. Van Leeuven et al. analyzed pre-operative [^68^Ga]Ga-PSMA-11 PET-CT and mpMRI in 140 patients with intermediate-to-high-risk prostate cancer [37]. In that study, the sensitivity (53%) and specificity (88%) of [^68^Ga]Ga-PSMA-11 PET-CT for predicting LNI were somewhat lower than in our cohort, but the values were greater than in our study for detecting SVI (sensitivity, 53%, specificity, 88%) [37]. Similar to our study, Kulkarni et al. reported that [^68^Ga]Ga-PSMA-11 PET-CT was superior to mpMRI at detecting LNI in intermediate-to-high-risk prostate cancer: for [^68^Ga]Ga-PSMA-11 PET-CT, the sensitivity was 80%, and the specificity was 90%, whereas, for mpMRI, the sensitivity was 44%, and the specificity was 79% [38]. Likewise, Petersen et al. reported that [^68^Ga]Ga-PSMA-11 PET-CT performed better than mpMRI in primary nodal staging [39]. In contrast to most studies, Yilmaz et al. reported an excellent performance by both mpMRI and [^68^Ga]Ga-PSMA-11 PET-CT for diagnosing LNI: the sensitivity was 100% for both mpMRI and PET, with a specificity of 100% for PET-CT and of 37% for mpMRI [40]. However, the study by Yilmaz et al. enrolled only 24 patients [40]. Zhang et al. also reported very high sensitivity and specificity (all values > 93%) for both [^68^Ga]Ga-PSMA-11 PET-CT and mpMRI for the primary detection of LNI in patients with intermediate-to-high-risk prostate cancer, which was partly due to the use of a 3T scanner, and a high proportion (>80%) of enlarged (>10 mm) metastatic nodes [41]. In contrast, Cytawa et al. reported a low sensitivity (35%) of [^68^Ga]Ga-PSMA-I&T PET-CT in diagnosing LNI, which was attributed to a high proportion of nodes with micrometastases and use of the [^68^Ga]Ga-PSMA-I&T tracer, which could have a lower affinity than the PSMA-11 tracer [42]. Although the results of previous studies vary considerably, most studies that used both mpMRI and [^68^Ga]Ga-PSMA-11 PET-CT reported similar results to our current results, with better performance by [^68^Ga]Ga-PSMA-11 PET-CT for N staging but worse performance for T staging. As is evident from the above work, [^68^Ga]Ga-PSMA-11 PET-CT can help identify lymph node metastases; however, due to suboptimal sensitivity, its utilization for clinical decision making has yet to be assessed.

In our cohort, the performance of the MSKCC nomogram and Partin tables in diagnosing LNI, SVI, and EPE was similar to that reported among 102 patients with high-risk prostate cancer [43], with the MSKCC nomogram performing better than Partin tables. The performance of the Briganti nomogram in our sample was similar to that in a study among nearly 20,000 patients with prostate cancer scheduled for radical prostatectomy (sensitivity, 90%; specificity, 46%) [44]. Similarly to our study, Hotker et al. observed that the Briganti nomogram performed better than mpMRI for predicting LNI (AUC, 0.89 vs. 0.73), with an AUC similar to that of [^68^Ga]Ga-PSMA-11 PET-CT (AUC, 0.82) [45]. The MSKCC, Partin tables, and the Briganti nomogram in our study had comparable AUCs to those of mpMRI and [^68^Ga]Ga-PSMA-11 PET-CT. However, for binary classifications, which are required for the individual patient, decision curve analyses showed that mpMRI offered a greater benefit than the MSKCC nomogram in detecting SVI, whereas [^68^Ga]Ga-PSMA-11 PET-CT was superior to this clinical score for detecting LNI.

One of the problems of analyzing the results is the human factor determining the recognition and name of a visible lesion. Currently, a lot of time is devoted to automatic image analysis, including through artificial intelligence algorithms. These allow the human factor to be reduced, for example, when recognizing and naming brain lesions [46]. There is also emerging work on the analysis of MRI images of the prostate allowing automatic segmentation and the determination of the extent of disease [47]. For PET-CT examinations, neural networks are able to classify physiological tracer accumulation in organs [48], and there is also emerging work on machine-learning-based analysis of quantitative [^18^F]DCFPyL PET-CT in the prediction of LNI and high-risk pathological tumor features in primary PCa patients [49].

Our study has some limitations. We used clinical nomograms to estimate the risk of LNI, SVI, and EPE, but optimal cut-offs were retrospectively calculated. Such an approach may overestimate the performance of nomograms. The study sample was small, and, therefore, larger studies are required. Additionally, in line with the PI-RADS guidelines, only one of the T2, DWI, and DCE sequences in the mpMRI protocol covered the entire pelvis. Using all three sequences could have improved the assessment of the regional prostatic lymph nodes. Moreover, ADC values may vary between the different scanners and types of post-processing software used in our study. Another limitation of our study is that the assessment of LNI relied on the qualitative decisions of the raters. However, currently, there is a lack of objective guidelines to quantitatively diagnose metastatic involvement. Future studies should employ additional imaging techniques to assess LNI. For example, dual-energy computed tomography may improve tissue characterization and the detection of subtle changes in lymph nodes [50].

## 5. Conclusions

The accurate primary T and N staging of prostate cancer is crucial to decide on pelvic nerve-sparing surgery and extended pelvic node dissection. This study among patients with intermediate-to-high-risk prostate cancer scheduled for radical prostatectomy showed that mpMRI was superior to [^68^Ga]Ga-PSMA-11 PET-CT for detecting SVI, but [^68^Ga]Ga-PSMA-11 PET-CT was better for predicting LNI. In general, the clinical nomogram scores were characterized by a greater sensitivity but a lower specificity for all the T stage and N stage outcomes compared with mpMRI or [^68^Ga]Ga-PSMA-11 PET-CT. The MSKCC nomogram performed the best of all the clinical scores. However, decision curve analyses showed a greater net benefit of [^68^Ga]Ga-PSMA-11 PET-CT and mpMRI compared with binary diagnoses based on the MSKCC nomogram for predicting LNI and SVI, respectively. We found it unfeasible to use [^68^Ga]Ga-PSMA-11 PET-CT to assess EPE, and the performance of [^68^Ga]Ga-PSMA-11 PET-CT in diagnosing SVI was poor. Thus, [^68^Ga]Ga-PSMA-11 PET-CT improved the N staging but not the T staging in patients with intermediate-to-high-risk prostate cancer scheduled for radical prostatectomy.

In conclusion, mpMRI and [^68^Ga]Ga-PSMA-11 PET-CT are complementary techniques to be used in conjunction for the primary T staging and N staging in patients with intermediate- and high-risk prostate cancer, which, in addition to knowledge of the stage of the disease, also provide data on the location of lesions, which is important for planning the optimal treatment for patients.

## Figures and Tables

**Figure 1 cancers-15-05838-f001:**
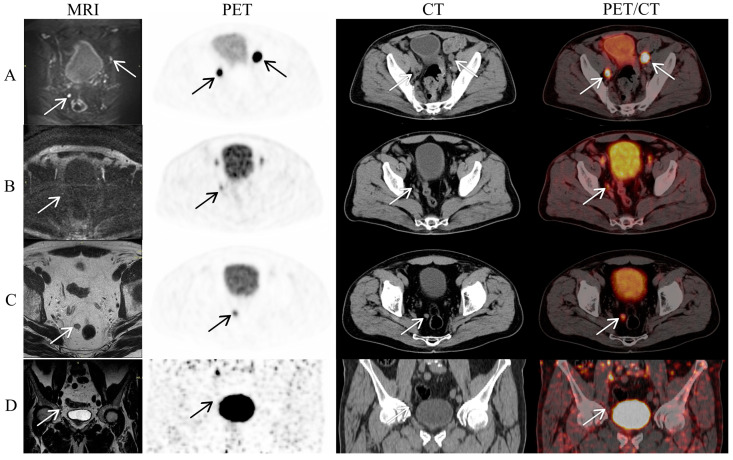
Lymph node involvement assessed on mpMRI and [^68^Ga]Ga-PSMA-11 PET-CT. (**A**) Arrows show pathologically confirmed metastatic lymph nodes that were correctly classified by both mpMRI and [^68^Ga]Ga-PSMA-11 PET-CT. (**B**) A pathologically confirmed metastatic lymph node missed by mpMRI but classified correctly by [^68^Ga]Ga-PSMA-11 PET-CT. (**C**) A non-metastatic lymph node that was incorrectly classified as metastatic by both mpMRI and [^68^Ga]Ga-PSMA-11 PET-CT. (**D**) A pathologically confirmed metastatic lymph node classified correctly by mpMRI but missed by [^68^Ga]Ga-PSMA-11 PET-CT.

**Figure 2 cancers-15-05838-f002:**
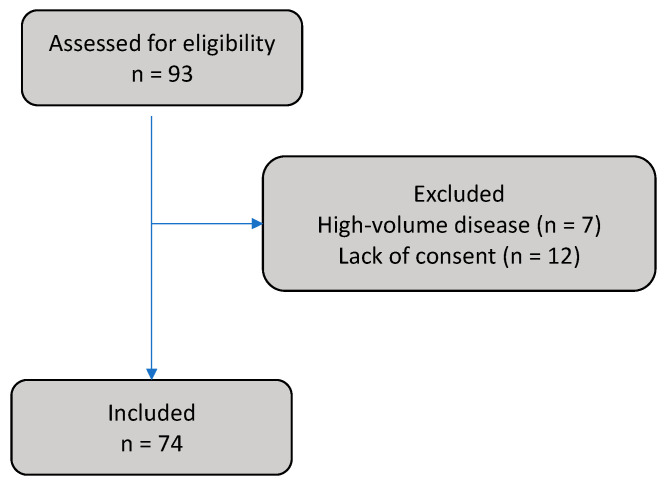
Patient selection.

**Figure 3 cancers-15-05838-f003:**
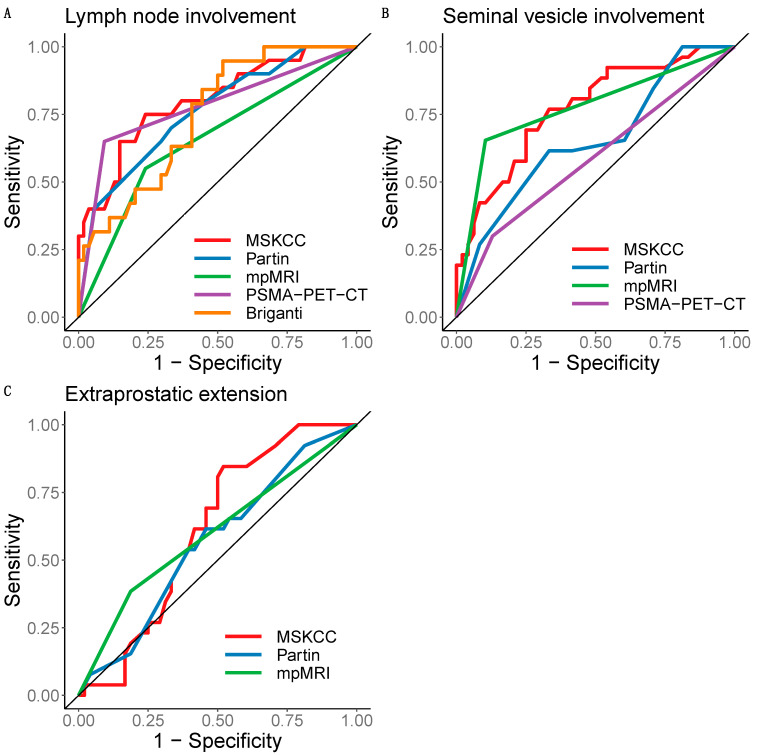
Receiver operating characteristic curves for detecting lymph node involvement (**A**), seminal vesicle involvement (**B**), and extraprostatic extension (**C**) by clinical scores and imaging method in patients with high-risk or intermediate-risk prostate cancer. MRI, multiparametric magnetic resonance imaging; PET-CT, positron emission computed tomography with prostate-specific membrane antigen as tracer; MSKCC, Memorial Sloan Kettering Cancer Center nomogram; Partin, Partin tables; Briganti, Briganti nomogram.

**Figure 4 cancers-15-05838-f004:**
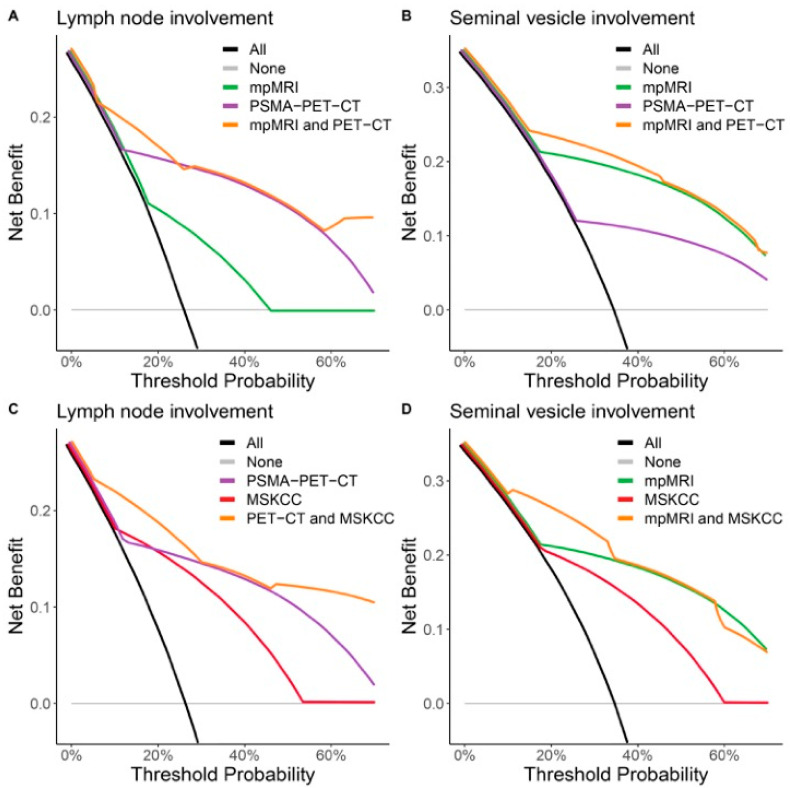
Decision curve analyses for lymph node involvement (**A**,**C**) and seminal vesicle involvement (**B**,**D**) in patients with high-risk or intermediate-risk prostate cancer. The *y*-axis is benefit, and the *x*-axis is preference (threshold probability). The benefit of a test or model is that it correctly identifies which patients do or do not have LNI or SVI. Threshold probability is defined as the minimum probability of an event at which a decision maker would take a given action. Net benefit is a weighted combination of true and false positives, where the weight is derived from the threshold probability. All, “diagnose in all patients”; none, “diagnose in none patients”; mpMRI, “diagnose by multiparametric magnetic resonance imaging”; PET-CT, “diagnose by positron emission computed tomography with prostate-specific membrane antigen as tracer”; MRI and PET-CT, “diagnose by both imaging methods”.

**Table 1 cancers-15-05838-t001:** Patients’ characteristics.

Characteristic	Total	Lymph Node Involvement on Pathology		*p*-Value
	*n* = 74 ^1^	No, *n* = 54 ^1^	Yes, *n* = 20 ^1^	
Age, years	66 (62, 71)	66 (62, 71)	67 (63, 69)	0.970 ^2^
PSA, before surgery (ng/mL)	13 (7, 28)	12 (7, 21)	24 (13, 40)	0.004 ^2^
DRE stage				0.074 ^3^
cT1c	25 (34%)	22 (41%)	3 (15%)	
cT2a	7 (9.5%)	4 (7.4%)	3 (15%)	
cT2b	16 (22%)	11 (20%)	5 (25%)	
cT2c	3 (4.1%)	3 (5.6%)	0 (0%)	
cT3a (EPE)	17 (23%)	12 (22%)	5 (25%)	
cT3b (SVI)	6 (8.1%)	2 (3.7%)	4 (20%)	
Risk group				0.015 ^3^
High	52 (70%)	36 (76%)	16 (100%)	
Intermediate	13 (18%)	13 (24%)	0 (0%)	
Metastatic	9 (12%)	5 (55%)	4 (45%)	
Prostate volume (mL)	42 (33, 54)	38 (30, 47)	50 (43, 56)	0.010 ^2^
PIRADS grade				0.010 ^3^
3	1 (1.4%)	1 (1.9%)	0 (0%)	
4	20 (27%)	19 (35%)	1 (5.0%)	
5	53 (72%)	34 (63%)	19 (95%)	
ISUP grade				<0.001 ^3^
2	6 (8.1%)	5 (9.3%)	1 (5.0%)	
3	24 (32%)	22 (41%)	2 (10%)	
4	32 (43%)	24 (44%)	8 (40%)	
5	12 (16%)	3 (5.6%)	9 (45%)	
Pathological T stage				<0.001 ^3^
pT2b	1 (1.4%)	1 (1.9%)	0 (0%)	
pT2c	21 (28%)	21 (39%)	0 (0%)	
pT3a	26 (35%)	21 (39%)	5 (25%)	
pT3b	26 (35%)	11 (20%)	15 (75%)	
LNI on pathology	20 (27%)			
T3a (EPE) on MRI	19 (26%)	15 (28%)	4 (20%)	0.496 ^4^
T3b (SVI) on MRI	22 (30%)	8 (15%)	14 (70%)	<0.001 ^4^
LNI on MRI	24 (32%)	13 (24%)	11 (55%)	0.012 ^4^
T3b (SVI) on PET-CT	13 (18%)	7 (13%)	6 (30%)	0.165 ^3^
LNI on PET	18 (24%)	5 (9.3%)	13 (65%)	<0.001 ^3^
Distant metastases on PET	9 (12%)	5 (9.3%)	4 (20%)	0.241 ^3^

^1^ Median (IQR); n (%), ^2^ Wilcoxon rank sum test, ^3^ Fisher’s exact test, ^4^ Pearson’s Chi-squared test. DRE, digital rectal examination; EPE, extra-prostatic extension; ISUP; International Society of Urological Pathology; LNI, lymph node involvement; PIRADS, Prostate Imaging Reporting and Data System; PSA, prostate-specific antigen; SVI, seminal vesicle involvement.

**Table 2 cancers-15-05838-t002:** Diagnostic performance of the MSKCC nomogram, Partin tables, mpMRI, [^68^Ga]Ga-PSMA-11 PET-CT, and Briganti nomogram for detecting lymph node involvement, seminal vesicle involvement, and extraprostatic extension in patients with high- or intermediate-risk prostate cancer.

Outcome	Method	Cut-Off	AUC	Sensitivity	Specificity	PPV	NPV
LNI	MSKCC nomogram	47.5	0.799 (0.680–0.918)	75.0 (55.0–90.0)	75.9 (64.8–87.0)	53.8 (41.4–68.2)	89.4 (81.8–95.8)
	Partin tables	22	0.761 (0.638–0.883)	70.0 (50.0–90.0)	66.7 (53.7–77.8)	43.8 (32.4–56.7)	85.7 (77.3–94.9)
	mpMRI	0.5	0.655 (0.529–0.780)	55.0 (35.0–75.0)	75.9 (64.8–87.0)	45.8 (31.8–63.2)	82.1 (75.0–90.0)
	[^68^Ga]Ga-PSMA-11 PET-CT	0.5	0.779 (0.665–0.893)	65.0 (40.0–85.0)	90.7 (81.5–98.1)	73.3 (56.0–92.3)	87.7 (81.0–94.3)
	Briganti nomogram	21	0.744 (0.624–0.864)	94.7 (84.2–100.0)	48.1 (35.2–61.1)	39.1 (33.3–47.4)	96.4 (88.5–100.0)
SVI	MSKCC nomogram	35.5	0.772 (0.659–0.885)	69.2 (50.0–88.5)	75.0 (62.5–85.4)	60.0 (46.7–73.9)	82.0 (72.5–91.7)
	Partin tables	21	0.654 (0.523–0.785)	61.5 (42.3–80.8)	66.7 (52.1–79.2)	50.0 (37.5–62.5)	76.2 (66.7–85.7)
	mpMRI	0.5	0.775 (0.672–0.878)	65.4 (46.2–84.6)	89.6 (79.2–97.9)	77.3 (61.9–93.8)	82.7 (75.0–90.9)
	[^68^Ga]Ga-PSMA-11 PET-CT	0.5	0.585 (0.473–0.698)	30.0 (10.0–50.0)	87.0 (77.8–94.4)	46.2 (21.0–72.7)	77.0 (71.9–82.8)
EPE	MSKCC nomogram	93.5	0.613 (0.486–0.740)	84.6 (69.2–96.2)	47.9 (33.3–62.5)	46.9 (39.1–55.6)	85.7 (72.4–96.6)
	Partin tables	38.5	0.566 (0.433–0.698)	61.5 (42.3–80.8)	54.2 (39.6–68.8)	42.4 (31.8–53.1)	72.4 (61.1–83.3)
	mpMRI	0.5	0.599 (0.488–0.709)	38.5 (19.2–57.7)	81.2 (70.8–91.7)	52.6 (35.0–72.7)	70.8 (64.8–78.4)

EPE, extra-prostatic extension; LNI, lymph node involvement; mpMRI, multiparametric magnetic resonance imaging; MSKCC, Memorial Sloan Kettering Cancer Center; NPV, negative prognostic value; PPV, positive prognostic value; SVI, seminal vesicle involvement.

## Data Availability

The data analyzed in this study are available upon request from the corresponding author.

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
