# Peer review of "Comparison of Multiparametric MRI, [68Ga]Ga-PSMA-11 PET-CT, and Clinical Nomograms for Primary T and N Staging of Intermediate-to-High-Risk Prostate Cancer"

_cancers, 2023, doi:10.3390/cancers15245838_

Round 1

Reviewer 1 Report

Comments and Suggestions for Authors

The submitted manuscript is devoted to the relatively interesting and hot topic of the application of diagnostic radiopharmaceuticals with the MRI modality. Accurate diagnosis and staging play a key role in further therapy, especially in follow-up radioligand therapy using 177Lu-PSMA. The manuscript is written clearly, comprehensibly and clearly. This is a retrospective study with a cohort of over 70 subjects after applying marginal conditions. The processing of the study is very interesting and I rate it as beneficial. The data presented in the study are sufficient and clearly presented. The results are sufficiently discussed and the literary background is extensive. I have a few formal comments on the work:
- using the 68Ga index, instead of 68Ga - I don't know if this is due to the font style, which is against IUPAC anyway
- two different font sizes are used in the abstract
- there should be a consistent writing of radiopharmaceuticals l.292 p.13 and l.60 p.2 in terms of [68Ga]-PSMA..., I do not assume the use of a carrier form of gallium here.
Also add a list of used abbreviations to the manuscript for clarity.

The manuscript clearly deserves a comprehensive conclusion and not a statement in two sentences, despite the extensive discussion. Therefore, I recommend significant additions to the conclusion.
Minor errors or criticisms do not reduce the quality of the submitted manuscript. Therefore, I recommend accepting it in the journal after performing minor revisions.

Author Response

Dear Reviewer, 

Reviewer 2 Report

Comments and Suggestions for Authors

Thank you for the opportunity to read this manuscript. In my opinion is well written. Paper is interesting and well prepared. Also collected results are presented in clear and friendly to reader manner. In my opinion Introduction and conclusion schould  be improved. Th Introduction is possible to add more current references.

Thank you  

Author Response

Dear Reviewer, 

Reviewer 3 Report

Comments and Suggestions for Authors

The proposed work is of great interest given that prostate cancer is the 2nd most common cancer in men worldwide, behind lung cancer.

The manuscript could be enriched by analyzing and citing the following segmentation-related works:

https://doi.org/10.1038/s41598-020-71080-0  

https://doi.org/10.3390/cancers14184399

The following conclusion 

"In conclusion, mpMRI and 68Ga-PSMA-11 PET-CT are complementary techniques to be used in conjunction for primary T staging and N staging in patients with intermediate and high-risk prostate cancer." 

will have to be extended to the numerical results obtained, as this one was predictable and therefore not original.

Comments on the Quality of English Language

This is correct technical English.

Author Response

Dear Reviewer, 

Reviewer 4 Report

Comments and Suggestions for Authors

The strehth of the study is in the prospective nature. 

My comments: 

Introduction: current practice guidelenes regarding indication of imaging in the initial prostate cancer staging should be mentioned 

Methods: 

The numbers of patients included do not fit. There were 9 M1 patients and 7 with high volume were excluded; that means 2 low volume M1 were included in the study. There is only IR and HR group in the Patients characteristics. Inclusion of low volume M1 patients into the HR group is not appropriate, please, correct. 

Pelvic nodes pathology specimen mapping and registration with imaging should be described in more details.

There are four tests cited for significance assessment of risk factors in Table 1. This is unsual and should be acknowledged/explained in 2.7 Stastistical analysis.

Results

Graphics of ROC curves look weird a more appropriate representation should be considered, the problem might be the low number of patients.

Figure 4

Decision curve analysis is diffucult to interprete for a reader, more detailed and clear comments should be included.

Discussion

lines 254 - 6: "We found unfeasible..." should be clarified with citation of literature or previous experience of authors. 

Lines 250 - 263: some citations are missing e. g. mpMRI is the best method.. or compared with previous work...

Lines 254 - 298 numbers of patients included in studies are not mentioned except of one study by Gupta, please add.

316-318 abbrevaitions DCE and ADC  should be explained.

There is little discussion on decision curve analysis findings must be put into context, please add.

Conclusion is correct but acknowledge (in the Discussion):

- PSMA PET/CT should not be used for T staging per guidelines

- PSMA PET/CT might be complementary to conventional imaging for N assessment but with suboptimal sensitivity and its utilisation for clinical decisions making has yet to be assessed.

Author Response

Dear Reviewer, 

Reviewer 5 Report

Comments and Suggestions for Authors

This study evaluated the role of MpMRI, PET PSMA, and nomograms for the primary staging of high, and intermediate prostate cancer patients. It is a well-structured study and the authors have addressed all the issues adequately. However, the study has serious limitations, such as the small size, the limitations on MRI, etc. Additionally, I believe that this is not an innovative study and may be a repetition of old studies describing the values of sensitivity and specificity of their study without anything new.  

Author Response

Dear Reviewer, 

Reviewer 6 Report

Comments and Suggestions for Authors

Dear Authors,

thank you for your interesting work concerning comparison of mpMRI and PSMA imaging for staging of high risk prostate cancer.

About the pour performance of PSMA imaging for EPE detection, have you an idea od the correlation with spatial resolution of your PET imager? Are you thinking that more resolved images (by improvement of the apparatuses or use of of 18F tracers) could improve theses performances ?

Sincerely yours 

Author Response

Dear Reviewer, 

Round 2

Reviewer 3 Report

Comments and Suggestions for Authors

The manuscript has been significantly improved. Therefore, it deserves to be published in Cancers.

Comments on the Quality of English Language

The English used is acceptable for a scientific article.

Reviewer 5 Report

Comments and Suggestions for Authors

Despite the effort from the authors to provide a better study, this study lacks novelty and so in my opinion this paper should be rejected